# Estimation of the Effect of Accelerating New Bone Formation of High and Low Molecular Weight Hyaluronic Acid Hybrid: An Animal Study

**DOI:** 10.3390/polym13111708

**Published:** 2021-05-24

**Authors:** Po-Jan Kuo, Hsiu-Ju Yen, Chi-Yu Lin, Hsuan-Yu Lai, Chun-Hung Chen, Shwu-Huey Wang, Wei-Jen Chang, Sheng-Yang Lee, Haw-Ming Huang

**Affiliations:** 1School of Dentistry, Department of Periodontology, National Defense Medical Center and Tri-Service General Hospital, Taipei 11490, Taiwan; kuopojan@gmail.com; 2Department of Dentistry, Division of Prosthodontics, Taipei Medical University Hospital, Taipei 11031, Taiwan; b202093069@tmu.edu.tw; 3School of Dentistry, College of Oral Medicine, Taipei Medical University, Taipei 11031, Taiwan; alexlin1018@gmail.com (C.-Y.L.); d204108002@tmu.edu.tw (H.-Y.L.); m8404006@tmu.edu.tw (W.-J.C.); seanlee@tmu.edu.tw (S.-Y.L.); 4Center for Tooth Bank and Dental Stem Cell Technology, Taipei Medical University, Taipei 11031, Taiwan; 5School of Biomedical Engineering, College of Medical Engineering, Taipei 11031, Taiwan; b812106027@tmu.edu.tw; 6Core Facility Center, Office of Research and Development, Taipei Medical University, Taipei 11031, Taiwan; shwu@tmu.edu.tw; 7Dental Department, Taipei Municipal Wanfang Hospital, Taipei 11031, Taiwan; 8Graduate Institute of Biomedical Optomechatronics, College of Biomedical Engineering, Taipei Medical University, Taipei 11031, Taiwan

**Keywords:** low molecular weight, hyaluronic acid, new bone formation, animal study

## Abstract

Osteoconduction is an important consideration for fabricating bio-active materials for bone regeneration. For years, hydroxyapatite and β-calcium triphosphate (β-TCP) have been used to develop bone grafts for treating bone defects. However, this material can be difficult to handle due to filling material sagging. High molecular weight hyaluronic acid (H-HA) can be used as a carrier to address this problem and improve operability. However, the effect of H-HA on bone formation is still controversial. In this study, low molecular weight hyaluronic acid (L-HA) was fabricated using gamma-ray irradiation. The viscoelastic properties and chemical structure of the fabricated hybrids were evaluated by a rheological analysis nuclear magnetic resonance (NMR) spectrum. The L-MH was mixed with H-HA to produce H-HA/L-HA hybrids at ratios of 80:20, 50:50 and 20:80 (*w*/*w*). These HA hybrids were then combined with hydroxyapatite and β-TCP to create a novel bone graft composite. For animal study, artificial bone defects were prepared in rabbit femurs. After 12 weeks of healing, the rabbits were scarified, and the healing statuses were observed and evaluated through micro-computer tomography (CT) and tissue histological images. Our viscoelastic analysis showed that an HA hybrid consisting 20% H-HA is sufficient to maintain elasticity; however, the addition of L-HA dramatically decreases the dynamic viscosity of the HA hybrid. Micro-CT images showed that the new bone formations in the rabbit femur defect model treated with 50% and 80% L-HA were 1.47 (*p* < 0.05) and 2.26 (*p* < 0.01) times higher than samples filled with HA free bone graft. In addition, a similar tendency was observed in the results of HE staining. These results lead us to suggest that the material with an H-HA/L-HA ratio of 50:50 exhibited acceptable viscosity and significant new bone formation. Thus, it is reasonable to suggest that it may be a potential candidate to serve as a supporting system for improving the operability of granular bone grafts and enhancing new bone formations.

## 1. Introduction

For years, bone tissue engineering has been faced with the challenge of reducing the amount of bone graft material used without affecting treatment efficiency. Indeed, a matrix carrier is necessary in some applications to maintain bone graft material without loss during an operation; however, the carrier used for bone graft argumentation will reduce the absolute amount of bone graft, which can result in a decrease in the effectiveness of the artificial material [1]. Thus, improving material handling ability while preserving bone graft effectiveness remains a challenge.

Hyaluronic acid (HA) is a polysaccharide consisting of alternating residues of d-glucuronic acid and *N*-acetylglucosamine [2] and, as a basic component of extracellular matrix, displays high biocompatibility [3]. Previous studies have reported that HA can be used in mixture with bone graft material to modify the surface of artificial bone and enhance bone cell migration and growth [4,5]. One animal study has shown that coating HA on the surface of titanium implants can increase bone formation efficiency at the implant/bone interface [2]. Several investigations have used HA as a carrier and when combined with various bone grafts for bone augmentation. Nguyen and Lee (2014), for example, fabricated a bone substitute consisting of HA-gelatin hydrogel and biphasic calcium phosphate [6], and found that this composite provided excellent cellular response and could enhance the mechanical strength of cancellous bones and increase their bearing ability when subjected to load. A similar phenomenon was reported in tests of a mixture of HA–based matrix and collagenated heterologous bone graft [7]. In previous studies, HA has been reported to be an excellent carrier for sinus augmentation without reducing the clinical effectiveness of the allograft [6,8].

However, the effect of HA on bone formation is controversial in the case of combining hydroxyapatite/β-calcium triphosphate (β-TCP) with HA. For example, while Elkarargy (2013) found the addition of HA increases osteoconduction efficiency compared to samples without HA [9], Aguado et al.’s report (2014) indicated that using linear hyaluronic acid did not result in healing of the grafted area, and the amount of formed bone was not significantly higher in samples with HA than with β-TCP granules alone [10]. This may be due to the major role of HA in the HA–hydroxyapatite/β-TCP system only being able to act as a structural enforcement of the surrounding environment [11].

The above investigations made use of high molecular weight HA (H-HA), which can act as a physical scaffold for cell migration but cannot act as a ligand to directly bind to receptors on cell surfaces [12]. Unlike the controversial effects of H-HA on cell proliferation and differentiation [13,14], low molecular weight HA (L-HA) is well known to provide positive effects for cell proliferation and differentiation [14,15,16]. In the initial stage of wound healing, HA with a molecular weight reaching 2000 kDa only accumulates in the extracellular matrix and combines with fibrinogen to form a clot. However, L-HA with molecular weight ranging from 80 to 800 kDa influences the inflammatory response and activates macrophages [17], resulting in accelerated wound repair [18,19]. Although previous in vitro cellular studies have reported that L-HA exhibits a positive effect on bone healing, the effect of L-HA combined with hydroxyapatite/β-TCP bone graft on the healing of bone defects is still unclear [15,20,21]. In addition, it has been reported that reducing HA molecular weight results in lowered viscosity, which may cause a loss of intended operational effect. Accordingly, we fabricated H-HA/L-HA hybrid composites at different mixing ratios to be used as a carrier to release L-HA and maintain hydroxyapatite/β-TCP bone grafts for this study, and evaluated their performances using an animal model.

## 2. Materials and Methods

### 2.1. Materials

Formaldehyde, xylazine, povidone iodine, hematoxylin, eosin regent and D_2_O were purchased from Sigma-Aldrich (Sigma-Aldrich Inc. St. Louis, MO, USA). Tiletamine-zolazepam (Zoletil 50) was obtained from Virbac (Virbac Co., Carros CEDEX, France). Hyaluronic acid was purchased from Cheng-Yi Chemical Industry Co. Ltd. Taipei, Taiwan. Decalcifier was purchased from Thermo Fisher Scientific, Inc., Cheshire, UK. Hydroxyapatite and β-TCP were commercial products from Wiltrom Ltd., Taipei, Taiwan.

### 2.2. Preparation of H-HA/L-HA Hybrids

The molecular weight of H-HA used in this study was measured at 3000 kDa. The L-HA was prepared according to a previous method [19]. Briefly, γ-rays from a cobalt 60 irradiator (Point Source, AECL, IR-79, Nordion, Ottawa, ON, Canada) were used to destroy the H-HA structure using a continuous 1 kGy/h dose of radiation at 22 °C for 20 h. After irradiation, the molecular weight of low-molecular-weight HA was 200 kDa [19]. HA hybrids were fabricated by mixing H-HA and prepared L-HA at ratios of 80:20, 50:50 and 20:80 (*w*/*w*). Briefly, 1 g of H-LA and L-HA mixture was added to 3.5 mL phosphate buffer solution (PBS). After gently stirring for 2 h, the hybrid was freeze-dried and stored in a moisture-proof environment for further use.

### 2.3. Physicochemical Properties Tests of the H-HA/L-HA Hybrid

The chemical structure of the H-HA/L-HA hybrids was evaluated using ^1^H nuclear magnetic resonance (NMR) spectrums obtained from a 500 MHz NMR spectrometer (DRX500 Avance, Bruker BioSpin GmbH, Rheinstetten, Germany). The measurements were performed at 27 °C. D_2_O was used as the solvent in all NMR experiments. The viscoelastic properties of the prepared HA hybrids were determined using a rheologic measurement device (Anton Paar MCR 302 rheometer, Anton Paar, Graz, Austria). The rheometer was equipped with a parallel plate with a plate diameter of 25 mm and gap of 1.0 mm at 25 °C. This device was calibrated according to a previous experiment [22]. Briefly, samples to be tested were prepared as a 10 mg/mL solution. At 30 min post-loading, the complex moduli (G*) of the HA hybrid were measured using a frequency sweep from 10 to 100 Hz. HA dynamic viscosity (η*) was recorded as a function of the shear rate range from 0.005 to 10 (1/s).

### 2.4. Animal Experiment

In this study, six New Zealand white rabbits (average weight 3.0–3.6 kg) were used as test samples. These white rabbits were raised in a stable environment at a temperature of 25 °C and a 50% humidity and provided with solid food and water. All animal procedures were performed according to the National Research Council’s Guide for the Care and Use of Laboratory Animals guidelines and protocols approved by the Institutional Animal Care and Use Committee of the National Defense Medical Center, Taipei, Taiwan (IACUC-17-236). Since the properties of 100% H-HA and L-HA were not the focus of this study and, according to comments from the committee and in acceptance of the guiding principles of Declaration of Helsinki, which supports a reduction in laboratory animal use, only H-HA/L-HA hybrids with mixing ratios of 80:20, 50:50 and 20:80 were tested in the following experiments.

Surgical procedures were performed under sterile conditions. Before surgery, general anesthesia was achieved with an intramuscular injection of tiletamine–zolazepam (Zoletil 50) at a dosage rate of 15 mg/kg. After each rabbit was deeply anesthetized, the surgical site was shaved and disinfected with povidone iodine, and the skin cut to expose the lateral femoral condyle. According to previous studies, cylindrical defects 5 mm in diameter and 10 mm in length were drilled in each rabbit’s left and right leg (Figure 1a–c) under saline cooling conditions [23,24]. The filling material was prepared by mixing 1.0 g prepared HA hybrid containing various concentrations of L-HA and 1.0 g Hydroxyapatite/β-TCP substitute (with a ratio of 60:40 *w*/*w*) (Figure 1d). For each artificial defect, 1.0 g of bone graft-HA hybrid was grafted (Figure 1e,f). After grafting, the periosteum was closed with absorbable inner and outer flap sutures (Figure 1g,h) (Vicryl^®^ 4.0, Ethicon, Somerville, NJ, USA). Postoperative antibiotics and analgesics were administered intramuscularly for three days. After 12 weeks of healing, the rabbits were euthanized by carbon dioxide asphyxiation under anesthesia (50 mg/mL Zoletil 50 at a dosage of 15 mg/kg). Bone tissues from the surgical site were collected and fixed in a 10% formaldehyde solution at pH 7.0. Artificial bone defect grafted with HA-free Hydroxyapatite/β-TCP substitute was set as the control. Three samples in each tested group were collected for further analysis.

### 2.5. Micro-CT Measurements

Femoral condyles containing the artificial defects were scanned using micro-computed tomography (micro-CT) (Skyscan 1076, SkyScan, Aartselaar, Belgium) with a 0.5-mm aluminum filter at an energy level of 75 kV and current of 200 μA, with a pixel resolution of 18 μA. Data were analyzed using quantifying reconstructed three-dimensional images. According to previous studies, the percentage of new bone formation in each defect was quantitated by calculating the ratio of the bone volume (BV) to the total tissue volume (TV) in the defect holes.

### 2.6. Histological and Histomorphometrical Evaluation

In order to observe changes in bone growth among the artificial defects treated with different filling samples, bone samples were evaluated using histological analyses. To decalcify the sample, bone blocks were immersed in decalcifier for 4 weeks. Samples were then dehydrated in alcohol with an increasing gradient concentration (60–100%), embedded in paraffin and cut into sections with a thickness of 5 μm. Finally, bone tissue specimens were stained with hematoxylin and eosin. Histological images were acquired using a microscope slide scanner (OPTIKA, Ponteranica, Italia). Areas of new bone formation, residual bone substitute and non-mineralized tissue in the defect were quantitatively analyzed using image analysis software (ImageJ, National Institutes of Health, Bethesda, MD, USA).

### 2.7. Statistical Analysis

Mean values and standard deviations of the percentages of newly formed bone, residual bone substitute and non-mineralized tissue quantified using micro-CT, and histological images were calculated and presented. Differences between samples with various amounts of L-HA were investigated using one-way analysis of variance (ANOVA) with Duncan’s post hoc test (SPSS Inc., Chicago, IL, USA). For all tests, statistical significance was defined as a *p* value less than 0.05.

## 3. Results and Discussions

### Characterization Results for Fe_3_O_4_ NPs

HA synthesized using common methodologies is of high molecular weight. To obtain L-HA of a specific molecular weight, both physical methods (ultrasound, ozone, electron beam, gamma-ray radiation and heat treatment) and chemical methods (enzyme and acid degradation) are used to destroy H-HA’s main structure [25,26,27,28]. However, when the main bonding chain of H-HA is destroyed, its viscoelasticity and water retention properties undergo significant changes. In this study, ^1^H NMR spectra were used to examine the chemical structure of the fabricated H-HA/L-HA hybrids. We found no observable changes in chemical structure among these hybrids at different H-HA/L-HA mixing ratios (Figure 2). These results are supported by a previous report which indicated that γ-ray irradiation preserves HA’s fundamental structure [19,29].

Viscoelasticity is an important property of HA that allows the use of HA in various medical applications. Our rheological tests showed that H-HA/L-HA with different mixing ratios has almost the same complex modulus value at frequencies higher than 10 Hz (Figure 3a), meaning that the H-HA/L-HA hybrids used in this study retained their elasticity at this frequency. This result is supported by a previous study which also reported that hybrid complexes created by mixing high and low molecular weight HA maintained their moduli and were suitable for the treatment of osteoarthritis [30]. Xue and coauthors (2020) also fabricated H-HA/L-HA hybrids to evaluate their potential applications in regenerative medicine and tissue engineering [31]. Their results showed that, when the mixing ratio of H-HA:L-HA reached 80:20, the moduli reduced dramatically. This may have been due to cross-linking of the H-HA and L-HA used in their experiment. In the present study, NMR spectra readings showed no observable chemical shift peaks at 1.5 and 1.8 (Figure 2), which provides evidence that the fabricated hybrids were non-cross-linked HA [31]. Complex modulus (root mean square of storage modulus and lost modulus) is a characteristic of the overall ability to resist deformation when a dynamic force is applied [22,32]. Results shown in Figure 3a indicate that 20% H-HA is sufficient to maintain the H-HA/L-HA hybrid’s elastic properties. Since bone tissues are subjected to a stress environment during daily life, the excellent elastic property shown in Figure 3a suggests that the H-HA/L-HA hybrid used in this study has good applicability in orthopedics.

However, a high proportion of H-HA reduces the osteoconductivity of H-HA/L-HA hybrids. CT analysis shows no difference in new bone formation (Figure 5a), amount of connective tissue (Figure 5b) or remaining material (Figure 5c) when artificial bone defects treated with bone graft containing the 80-20 HA hybrid were compared to the bone graft only group. This result consistent with a previous report that indicated the major role of H-HA in graft material is to maintain a stable shape rather than directly affect bone regeneration [10]. However, quantitative results show that bone defects treated with bone graft and L-HA/H-HA hybrids containing higher L-HA amounts exhibit better bone reparative processes (Figure 4). The new bone formation of bone defects treated with the bone graft-HA hybrid complex with 50% and 80% L-HA in L-HA/H-HA hybrids significantly increased new bone formation (Figure 5a) and decreased non-mineralized tissue (Figure 5c). New bone formation in bone defects treated with 50 H-HA/50 LHA and 80 H-HA/20 L-HA hybrid complexes were 1.47 (*p* < 0.05) and 2.26 (*p* < 0.05) times higher than defects treated with HA-free bone graft (Figure 5a). In addition, non-mineralized tissue of these filling materials was decreased 1.62 fold for 50 H-HA/50 L-HA and 1.21 fold for 80 H-HA/20 L-HA hybrids (*p* < 0.05) (Figure 5c). A similar conclusion can also be reached from histological assessment (Figure 6).

Quantification of histological images showed new bone formation in bone defects treated with 50 H-HA/50 LHA and 80 H-HA/20 L-HA hybrid complexes at 31.21 ± 4.72% and 42.00 ± 4.72%, respectively. These values are significantly higher (*p* < 0.05) than samples filled with HA-free bone graft (19.67 ± 5.21%) (Figure 7a). The residual bone substitute (33.38 ± 3.31%) and non-mineralized tissue (25.38 ± 3.21%) of the bone graft mixed with 80 H-HA/20 L-HA hybrids were also significantly decreased compared to the HA-free bone graft group (*p* < 0.05) (Figure 7b,c). These histological analyses also indicate that artificial bone defects filled with more L-HA results in a better bone healing process. Combined with the CT results (Figure 4 and Figure 5) and histomorphometric analysis (Figure 6 and Figure 7), it is r0easonable to conclude that the effect of HA on bone healing is strongly affected by its molecular weight. These results confirm that L-HA provides an osteo-regenerative effect on bone reparation. Aguado et al. (2014) also investigated the use of HA as an aqueous binder of β-TCP granules [10]. After implanting an HA/β-TCP composite mixture into artificial holes drilled in the femoral condyle of rabbit legs, they also found that β-TCP granules mixed with HA induce an increase in bone apposition. They concluded that HA’s role appears to be as a vehicle only as it does not interfere with bone remodeling; however, this is because only H-HA was used in their study. The effects of high molecular weight HA on cell proliferation and differentiation remain controversial [13,33].

The prepared 80 H-HA/20 L-HA hybrid shows a typical viscosity curve (Figure 3b). The dynamic viscosity of this HA hybrid depends on the shear rate as a non-Newtonian liquid at 0.005–10 s^−1^ [34]. When the amount of L-HA was increased to 80%, a shear thinning phenomenon occurred as seen in the dramatically decreased slope (Figure 3b). When the strain rate reached 2.8 s^−1^, the viscosity of this 20 H-80 L HA hybrid increased with the increasing strain rate and exhibited shear-thickening. This hardening phenomenon can provide a shock-damping function and protective effect when a sudden high-load impact is applied [19]. Although the bone graft with 20 H-HA/80 L-HA hybrid demonstrated almost double the new bone formation compared to the HA-free sample (Figure 5a and Figure 7a), the sharp decrease in viscosity (Figure 3b) reduced its adhesive properties and may limit the applications of this material. For example, guided bone regeneration (GBR) is an important dental surgery that regenerates bone mass and increases bone width for healing after artificial dental implant procedures. For years, the most commonly used material for GBR has been a mixture of hydroxyapatite and β-calcium triphosphate (β-TCP) [35,36,37]. However, bone grafts composed of these two particles only are difficult to handle at posterior maxilla due to sagging of the filling material [38,39]. Therefore, maintaining adequate viscosity for operability and retaining the osteo-regenerative efficiency of fabricated bone graft composites has become an issue for investigation. It has been reported that when H-HA and L-HA are mixed at a ratio of 1:1, their viscosity and in vitro resistance to enzymatic hydrolysis can be improved [31,40]. Interestingly, combined data seen in Figure 3b, Figure 4c and Figure 6c indicate that the fabricated bone graft composite containing the 50/50 (*w*/*w*) H-HA/L-HA hybrid exhibited acceptable viscosity and significant new bone formation. This material may be a potential candidate for sinus lift augmentation of the posterior maxilla.

The most important finding of this study is a confirmation that L-HA can promote new bone formation and decrease the percentage of non-mineralized tissue during the healing process (Figure 4, Figure 5, Figure 6 and Figure 7). The mechanism of this osteo-regenerative effect was not assessed in this study but has been reported by several investigations which indicate that the effect of smaller hyaluronan molecules (at molecular weights ranging from 5 to 20 kDa) on regeneration is due to inducing cytokine and inflammatory responses at an early stage [17,41]. However, an inflammatory response was not observed when current bone graft composites were implanted into the bone defect (Figure 6), which may be due to the molecular weight of L-HA used in this study being about 200 kDa [19]. For HA with molecular weights in the 50–200 kDa range, tissue regeneration effects due to stem cell differentiation [21] and cell proliferation [16] were reported. Ariyoshi et al. (2005) indicated that L-HA promotes bone tissue engineering through enhancing the interaction of RANKL and RANK, which activates the signal transduction pathway involved in osteoclast differentiation [20]. However, besides the biological effects mentioned above, physical factors should also be taken into account. In this study, a new hypothesis was proposed to explain the positive effect of H-HA/L-HA hybrids on bone healing based on physical properties. The high elasticity (Figure 3a) of the current H-HA/L-HA hybrid allows it to exhibit a high level of resistance to hyaluronidase [17,30,31,42]. Thus, bone graft composites with an L-HA proportion above 50% provide a niche to prolong the viscosupplementation and bioactive effects in tissue engineering. In addition, low viscosity allows bone graft composites to exhibit two other benefits for bone regeneration. The first benefit is that L-HA can easily be released from the hybrid and allow bone regeneration. As a high viscosity environment may hamper cell movement, the second benefit is that reducing viscosity while maintaining elasticity the H-HA/L-HA hybrid can change the cell-area mechanical environment and allow cells increased mobility [17].

In conclusion, a novel composite mixed with H-HA/L-HA hybrid and hydroxyapatite/β-TCP can improve the osteoconductive properties of bone grafts. The results present in this study suggest that a 1:1 H-HA/L-HA hybrid mixture can serve as a supporting system to improve the operability of bone grafts and enhance new bone formation.

## Figures and Tables

**Figure 1 polymers-13-01708-f001:**
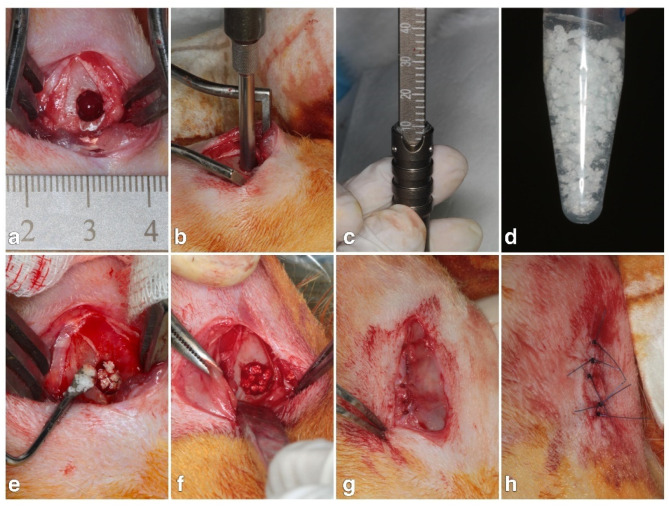
(**a**) Surgical incision and exposure of lateral femoral condyle. (**b**,**c**) Bony defects 5 mm in diameter and 10 mm in length were created in both femurs using a surgical drill. (**d**) Bone substitute mixed with hyaluronic acid hybrid. (**e**,**f**) Grafting material filled into the defect site. (**g**) Inner suture to close periosteum. (**h**) Outer flap suture for primary wound closure.

**Figure 2 polymers-13-01708-f002:**
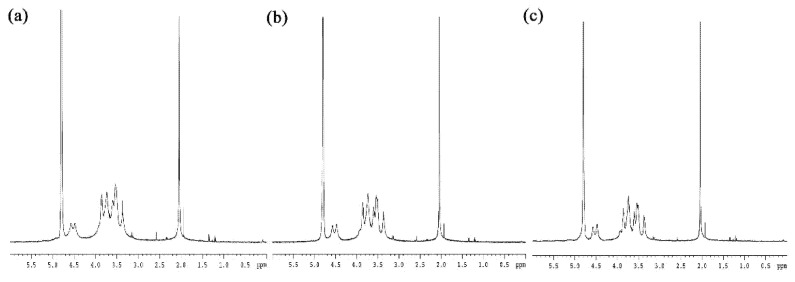
^1^H NMR spectra of H-HA/L-HA hybrids at mixing ratios of (**a**) 80:20, (**b**) 50:50 and (**c**) 20:80.

**Figure 3 polymers-13-01708-f003:**
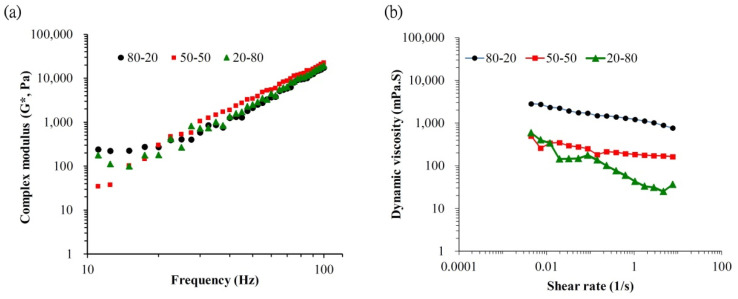
(**a**) Mechanical spectra of the complex modulus (G*) and (**b**) dynamic viscosity (η*) of the H-HA/L-HA hybrids prepared in this study.

**Figure 4 polymers-13-01708-f004:**
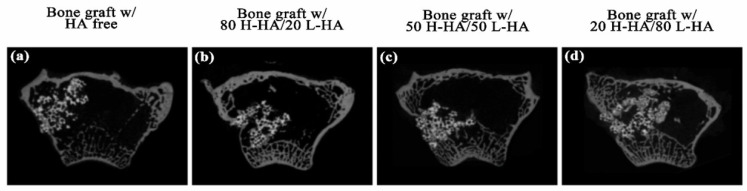
Micro-CT transversal section images of the tested artificial defects grafted with (**a**) HA-free hydroxyapatite/β-TCP bone graft and bone graft combined with HA hybrid at H-HL/L-HA ratios of 80:20 (**b**), 50:50 (**c**) and 20:80 (**d**) after 12 weeks of healing.

**Figure 5 polymers-13-01708-f005:**
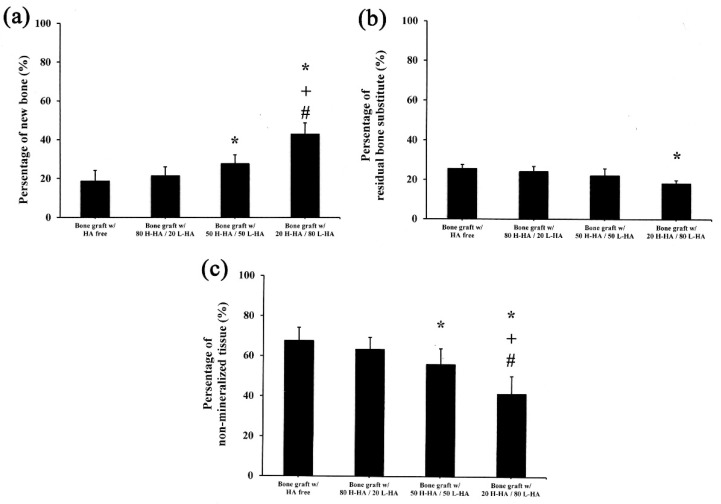
Statistical analysis of micro-CT quantification of (**a**) new bone formation, (**b**) residual bone substitute and (**c**) non-mineralized tissue. Data are expressed as mean ± standard deviation. The symbol * denotes a significant difference from the “bone graft with HA free” group, + denotes a significant difference from the “bone graft with 80 H-HA/20 L-HA” group, # denotes a significant difference from the “bone graft with 50 H-HA/50 L-HA” group. Significance (*p* < 0.05) was determined using the post hoc Duncan test after one-way ANOVA.

**Figure 6 polymers-13-01708-f006:**
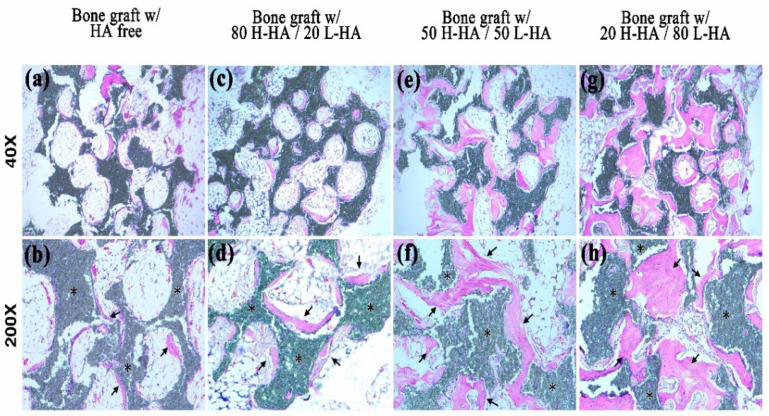
Histological images of tissue sections at artificial defects in rabbit femoral condyles. Microscopy images of bone graft with HA free (**a**,**b**), bone graft with 80 H-HA/20 L-HA (**c**,**d**), bone graft with 50 H-HA/50 L-HA (**e**,**f**) and bone graft with 20 H-HA/80 L-HA groups (**g**,**h**). The symbol * denotes the residual bone graft. Black arrows indicate newly formed bone. (upper panel: 40×; lower panel: 200×).

**Figure 7 polymers-13-01708-f007:**
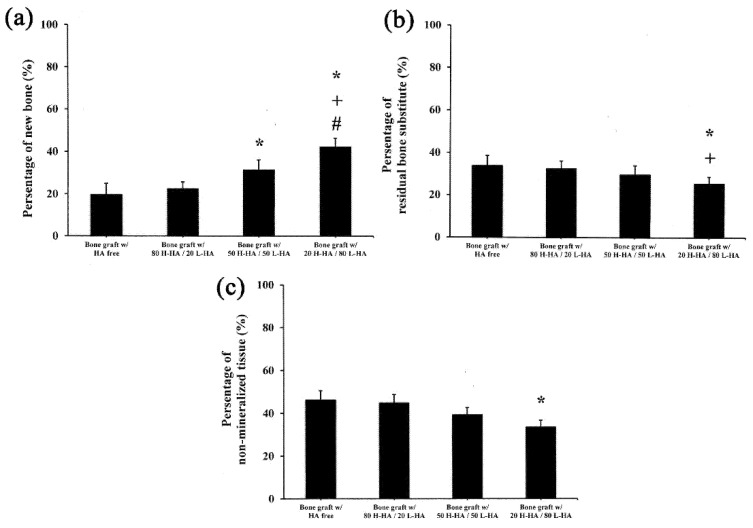
Statistical analysis of histological quantification of (**a**) new bone formation, (**b**) residual bone substitute and (**c**) non-mineralized tissue. Data are expressed as mean ± standard deviation. Symbols *, + and # denote significant differences (*p* < 0.05) from the bone graft with HA free, bone graft with 80 H-HA/20 L-HA and bone graft with 50 H-HA/50 L-HA groups, respectively. Significance was determined using the post hoc Duncan test after one-way ANOVA.

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
