# Peer review of "Estimation of the Effect of Accelerating New Bone Formation of High and Low Molecular Weight Hyaluronic Acid Hybrid: An Animal Study"

_polymers, 2021, doi:10.3390/polym13111708_

Round 1
Reviewer 1 Report
The scientific paper "Estimation of the effect of accelerating new bone formation of high and low molecular weight hyaluronic acid hybrid: An animal study" present results that suggest that the material with a H-HA/L-HA ratio of 50:50 exhibited acceptable viscosity and significant new bone formation. It can be considered that:
1) In the title, I suggest removing “An animal study”
2) The Abstract should reduce the introduction and justification part and increase the methodology, which is poor. Remove the repetition “In this study”.
3) In the introduction, line 77, insert the meaning of β-TCP = β-calcium triphosphate.
4) Complement the information of commercial products such as Zoletil 50 (manufacturer, city, country).
5) Line 142, put figure number (1)
6) Figure 5 presents difficulties for the readers to identify the groups (practically incomprehensible).
7) There is an excessive use of the term “in addition” throughout the entire manuscript. Please adjust.
Author Response
- In the title, I suggest removing “An animal study”
Author response: We thank all the comments from the reviewer. In title, the phrase “An animal study” was removed.
- The Abstract should reduce the introduction and justification part and increase the methodology, which is poor. Remove the repetition “In this study”.
Author response: In the revised manuscript, the abstract was rewritten. The instruction part was reduced and the methodology part was increased. In addition, the phrase “in this study” was removed throughout the abstract.
- In the introduction, line 77, insert the meaning of β-TCP = β-calcium triphosphate.
Author response: The meaning of β-calcium triphosphate (β-TCP) was added to line 70.
- Complement the information of commercial products such as Zoletil 50 (manufacturer, city, country).
Author response: The commercial information of Zoletil 50 was listed on lines 95-96.
- Line 142, put figure number (1)
Author response: The missing typo of Figure number (1) was added to line 137.
- Figure 5 presents difficulties for the readers to identify the groups (practically incomprehensible).
Author response: Figures 5 and 7 were enlarged to help the reader to identify the groups.
- There is an excessive use of the term “in addition” throughout the entire manuscript. Please adjust.
Author response: The excessive used of “in addition” was revised and replaced by other phrases throughout the paper.

Reviewer 2 Report
The manuscript “Estimation of the effect of accelerating new bone formation of high and low molecular weight hyaluronic acid hybrid: An animal study” deals with the production of a low molecular weight hyaluronic acid (L-HA) by gamma-ray irradiation for bone regeneration. Intriguing results have been obtained using this material to repair rabbit femurs. The publication is recommended; but after some revisions.
Detailed comments:
- Introduction. The state of the art related to bone regeneration can be enlarged; for instance, see this review: Baldino et al., Regeneration techniques for bone-To-Tendon and muscle-To-Tendon interfaces reconstruction, British Medical Bulletin, 2016, 117, pp. 25-37.
- The main chemico-physical properties of the materials used have to be added to section 2.1.
- The preparation of HA hybrids should be clarified. Moreover, the possible scalability of the method to prepare L-HA should be addressed.
- A comparison with the commercial HA should be performed.
- The Conclusions paragraph has to be added.
Author Response
- The manuscript “Estimation of the effect of accelerating new bone formation of high and low molecular weight hyaluronic acid hybrid: An animal study” deals with the production of a low molecular weight hyaluronic acid (L-HA) by gamma-ray irradiation for bone regeneration. Intriguing results have been obtained using this material to repair rabbit The publication is recommended; but after some revisions.
Author response: We thank the comments from the reviewer.
- The state of the art related to bone regeneration can be enlarged; for instance, see this review: Baldino et al., Regeneration techniques for bone-To-Tendon and muscle-To-Tendon interfaces reconstruction, British Medical Bulletin, 2016, 117, pp. 25-37.
Author response: We thank this comment from the reviewer. In Lines 192-193, a new introduction was added to enlarge the state of tissue engineering related to bone-To-Tendon and muscle-To-Tendon. Baldino et al.’s work was added as reference #29 in the revised manuscript.
- The main chemico-physical properties of the materials used have to be added to section 2.1.
Author response: All the materials and chemicals used for chemico-physical property tests were listed in section 2.1
- The preparation of HA hybrids should be clarified. Moreover, the possible scalability of the method to prepare L-HA should be addressed.
Author response: We thank the comments from the reviewer. The prepared method of HA hybrid was added to lines 106-108. The possible method for preparing L-HA was presented in lines 180-182.
- A comparison with the commercial HA should be performed.
Author response: We thank the comments from the reviewer. As our limited knowledge, until now, there is no commercially available product of H-HA/L-HA hybrid.
- The Conclusions paragraph has to be added.
Author response: A conclusion section was added to lines 313-317.

Round 2
Reviewer 2 Report
The authors perfomed the modifications proposed by the Reviewer and improved the manuscript. The publication is recommended.